# Detection of Metabolic Syndrome Using Insulin Resistance Indexes: A Cross-Sectional Observational Cohort Study

Lucas Fornari Laurindo [1,2], Giulia Minniti [1], Ricardo José Tofano [1], Karina Quesada [1], Eduardo Federighi Baisi Chagas [3,4] and Sandra Maria Barbalho [1,4,5,*]

[1] Department of Biochemistry and Pharmacology, School of Medicine, University of Marília (UNIMAR), Avenida Hygino Muzzy Filho, 1001, Marília 17525-902, SP, Brazil; lucasffffor@gmail.com (L.F.L.); giulia.minniti@hotmail.com (G.M.); rtofano@uol.com.br (R.J.T.); karinarquesada@gmail.com (K.Q.)

[2] Department of Biochemistry and Pharmacology, School of Medicine, Faculdade de Medicina de Marília (FAMEMA), Avenida Monte Carmelo, 800, Marília 17519-030, SP, Brazil

[3] Postgraduate Program in Health and Aging, Faculdade de Medicina de Marília (FAMEMA), Avenida Monte Carmelo, 800, Marília 17519-030, SP, Brazil; efbchagas@unimar.br

[4] Postgraduate Program in Structural and Functional Interactions in Rehabilitation, University of Marília (UNIMAR), Avenida Hygino Muzzy Filho, 1001, Marília 17525-902, SP, Brazil

[5] Department of Biochemistry and Nutrition, School of Food and Technology of Marília (FATEC), Avenida Castro Alves, 62, Marília 17500-000, SP, Brazil

* Correspondence: smbarbalho@gmail.com

**Abstract:** Insulin resistance (IR) is considered cardinal to the pathophysiology of metabolic syndrome (MetS). Previously, several simple indexes of IR calculated from biochemical and anthropometric variables have been proposed. However, these indexes are population-dependent; therefore, further studies on a global scale are necessary. The present study assessed the diagnostic accuracy of eight IR indicators, namely, METS-IR, TG-HDL-c, TyG, TyG-BMI, TyG-NC, TyG-NHtR, TyG-WC, and TyG-WHtR, in indicating MetS among a Brazilian population. For this, 268 patients (152 men and 116 women, 53–59 years of age) were included in the study, out of which 111 were diagnosed with MetS according to the National Cholesterol Education Program Adult Treatment Panel III (NCEP ATP III). All indexes achieved significant accuracy, with TyG-WC (0.849 (0.800–0.889)), TyG (0.837 (0.787–0.879)), and TG-HDL-c (0.817 (0.765–0.861)) having the highest area under the curve (AUC). Further, the most heightened diagnostic sensitivities were observed for TG-HDL-c (90.99%), TyG-WC (89.19%), and TyG-NC (84.68%), whereas the highest diagnostic specificities were noted for TyG (73.89%), TyG-WHtR (72.61%), and TyG-WC (66.88%). Thus, TyG-WC, TyG, and TG-HDL-c reached the greatest AUC values in our analyses, making them useful diagnostic indicators of MetS, and crucial for patients' clinical management.

**Keywords:** metabolic syndrome; Brazilian population; clinical indicators; insulin resistance; diabetes; METS-IR; TG-HDL-c; TyG; TyG-BMI; TyG-NC; TyG-NHtR; TyG-WC; TyG-WHtR

## 1. Introduction

Lifestyle changes are considered a cause of many diseases, including metabolic syndrome (MetS). This cardiometabolic condition is significantly related to an increase in cardiovascular risk, which can lead to cardiovascular diseases (CVDs) and other potentially fatal health outcomes, along with chronic and degenerative disorders associated with high morbidity and mortality rates [1–5].

MetS is a chronic, multifactorial, degenerative, and non-infective complex syndrome that is considered a global epidemic. Clinically, it is characterized by different cardiometabolic risk factors such as hypertension, dyslipidemia, impaired glucose metabolism, and abdominal obesity. Pathologically, MetS triggers a chronic low-grade inflammatory state and meta-inflammation derived from the inflamed fatty tissue that is directly related to CVDs, and

it affects pre-existing diseases such as arthritis and non-alcoholic fatty liver disease. Aside from the pro-inflammatory state, MetS is also associated with a pro-oxidative condition in which reactive oxygen species overload the antioxidant systems, causing DNA and post-translational protein alterations as well as lipid peroxidation, which, in turn, makes lipids more atherogenic [6–9]. According to the available literature, the most significant contributor to the association between metabolic syndrome and oxidative stress appears to be obesity and insulin resistance (IR), both of which are vital components of metabolic syndrome [10].

Globally, atherosclerosis is responsible for the majority of CVD-related morbidity and mortality, and it is strongly influenced by risk factors such as type 2 diabetes and MetS. The primary CVDs associated with atherosclerosis are ischemic heart disease, cerebrovascular disease, and peripheral arterial disease. Although the treatment of atherosclerotic CVD often involves angioplasty and stenting, which can be highly effective, restenosis, i.e., the re-narrowing of the artery after treatment, can significantly reduce its efficacy and, therefore, may necessitate re-intervention [11,12].

Many criteria have been formulated to help diagnose MetS; however, increasing evidence suggests that this metabolic condition is greater than the sum of its individual risk factors. For instance, the role of adipokines has been recognized with the ectopic fat being central to a MetS diagnosis. Adipokines, such as adiponectin and leptin, affect the whole body endocrinologically. Additionally, several genes that link the occurrence of MetS with homogeneous populations have also been identified [13,14].

One can define IR as a state of decreased tissue sensitivity to insulin. It is considered cardinal to the pathophysiology of MetS. The gold-standard method to measure insulin sensitivity is the hyperinsulinemic euglycemic clamp (HEC), which consists of continuous and concomitant insulin and glucose infusion until a serum insulin concentration of 100 mIU/L is achieved and maintained. Moreover, during exogenous hyperinsulinemia, there is a reduction in pancreatic and hepatic glucose production caused by an increase in glucose disposal into tissues. Furthermore, the hepatic glucose production as the amount of glucose administered reflects the tissues' uptake and, thus, the insulin sensitivity directly. However, HEC is expensive as well as time- and resource-consuming. Due to all these reasons, the IR state is most often assessed through simple clinical indicators called IR indexes [15,16]. Previous studies investigated the use of IR indicators in the diagnostic pathway of MetS and diabetes, but given that IR clinical indicators are usually population-dependent, there is a need to conduct further studies for each population.

IR indexes are tools that clinicians all around the world can use to identify metabolic alterations throughout patient charts. However, despite the existing literature on the clinical importance of MetS and IR indexes, little is known about the reliability of the indexes in detecting MetS among Brazilian individuals. Besides, thus far, no consensus has been reached on which index is the best. Therefore, this study aimed to compare the diagnostic accuracy of a group of newly proposed IR indexes that only consider anthropometric and simple biochemical measurements by applying them to the analysis of a cohort of patients from the Midwest region of Brazil.

## 2. Materials and Methods

### 2.1. Study Design

The participants in this cross-sectional observational cohort study were 268 patients who attended the University Hospital of the University of Marília (Marília, SP, Brazil) as well as visited in the cardiology unit of the University Hospital for routine cardiovascular consultations. This study was carried out between June–December 2021.

### 2.2. Study Population

This study was conducted on 268 adult and elderly volunteers who were over the age of 20 and consisted of 152 men and 116 women. During the study, 111 (70 men and 41 women) participants were diagnosed with MetS according to the National Cholesterol Education Program Adult Treatment Panel III (NCEP ATP III) criteria. The maximum

interval between the collection of biochemical tests and the performance of the physical examination to collect the anthropometric data was three months for all the patients included in the study.

The inclusion criteria comprised all patients of both sexes who attended the University Hospital for consultations as well as the cardiology unit of the University Hospital, were not previously diagnosed with MetS, and were over the age of 20. Conversely, the exclusion criteria comprised pregnant and lactating women and patients with a prescription for continuous treatment with any lipid-lowering, glycemic-lowering, insulin-sensitizing, or antihypertensive drugs.

*2.3. Anthropometric Data*

The following anthropometric parameters were investigated: height (in cm), weight (in kg), waist circumference (WC, in cm), and neck circumference (NC, in cm). Further, the body mass index of each participant was calculated based on their height and weight. The trainees of the University of Marília's nutrition and medicine schools, with previous training based on the techniques preconized by Lohman et al. [17] and Gibson [18], obtained the anthropometric measurements of each participant. Each participant's anthropometric measurements were taken in the morning. For this, the participants were asked to fast for at least 12 h after their last meal.

The height of each participant was measured in an upright standing position after removing shoes, with an accuracy of 0.5 cm. The size of the WC was measured at the approximate midpoint between the top of the iliac crest and the lower margin of the last palpable rib with the help of an unstretchable tape (in cm), while NC was measured at the midpoint of the neck between the mid-cervical spine and the mid-anterior neck, with an accuracy of 0.5 cm. Further, the body mass was obtained by positioning each participant, without shoes and only in their underwear, on an electronic weighing scale, with an accuracy of 0.1 kg. The waist-to-height ratio (WHtR) is calculated as waist circumference divided by height, whereas the neck-to-height ratio (NHtR) is calculated as neck circumference divided by height.

*2.4. Blood Pressure Measurements*

Blood pressure (BP) measurements were carried out by trainees of the University of Marília's nutrition and medicine schools by using the measurement techniques preconized by Rushton and Smith [19].

*2.5. Laboratory Analyses*

Biochemical analyses were conducted in accordance with the São Francisco Laboratory protocols of the University Hospital of the University of Marília. The São Francisco Laboratory uses the preconized reference values of the manufacturer of each biochemical test to obtain the results in the analyses. The following biochemical parameters were investigated: fasting total triglycerides (TG), fasting blood glucose (FBG), and high-density lipoprotein cholesterol (HDL-c).

*2.6. Diagnosis of Metabolic Syndrome*

MetS was defined based on the criteria established by the NCEP ATP III. According to this organization, MetS is determined by the occurrence of alteration among at least three of the following five components: (i) fasting blood glucose (FBG) $\geq$ 100 mg/dL or pharmacological treatment, (ii) systolic blood pressure (SBP) $\geq$ 130 mmHg or diastolic blood pressure (DBP) $\geq$ 85 mmHg or taking antihypertensive medication, (iii) triglycerides $\geq$ 150 mg/dL or pharmacological therapy, (iv) HDL-c $\leq$ 40 mg/dL for men and $\leq$50 mg/dL for women, and (v) waist circumference $\geq$ 102 cm for men and $\geq$88 cm for women [20,21]. Assessing waist circumference based on race/ethnicity is the predominant approach for defining MetS, given the variation in visceral adipose tissue and associated cardiometabolic

risk among people of different races and ethnicities. Nevertheless, defining race/ethnicity can present challenges in the clinical setting [22].

*2.7. Calculation of Body Mass Index*

The BMI values were reported according to the following classifications by the World Health Organization (WHO): underweight (<18.5 kg/m$^2$), average weight (18.5–24.9 kg/m$^2$), overweight (25–29.9 kg/m$^2$), and obese ($\geq$30 kg/m$^2$). This anthropometric variable can be calculated as weight (in km) divided by height (in m$^2$) [23].

*2.8. Calculation of Clinical Indicators*

The IR clinical indicators were calculated by using the formulas described by Mirr et al. [15]. These indicators and their calculations are as follows: (i) triglyceride-glucose index (TyG) = Ln (fasting TG [mg/dL] $\times$ FBG [mg/dL]/2), (ii) triglycerides to high-density lipoprotein cholesterol index (TG/HDL-c) = fasting TG (mg/dL)/fasting HDL cholesterol (mg/dL), (iii) the metabolic score for IR (METS-IR) = Ln [(2 $\times$ FBG (mg/dL) + fasting TG (mg/dL)] $\times$ BMI (kg/m$^2$))/(Ln [HDLc (mg/dL)], (iv) triglyceride glucose index-waist to height ratio (TyG-WHtR) = TyG $\times$ WHtR, (v) triglyceride glucose index body mass-index (TyG-BMI) = TyG $\times$ BMI, (vi) triglyceride glucose index-waist circumference (TyG-WC) = TyG $\times$ WC, (vii) triglyceride glucose index-neck circumference (TyG-NC) = TyG $\times$ NC, and (viii) triglyceride glucose index-neck circumference to height ratio (TyG-NHtR) = TyG $\times$ NHtR.

*2.9. Ethics Approval and Consent to Participate*

The protocols of this study were approved on 30 May 2021, by the Ethics Committee of the University of Marília under ethical approval number 50817221.6.0000.5496. The study was initiated only after all participants signed the free informed consent forms under Resolutions 466/2012 and 510/2016 of the Brazilian National Health Council. All the procedures in the study met the ethical standards outlined by the university's Institutional Ethics Committee and the Declaration of Helsinki (revised in 2008).

*2.10. Statistical Analyses*

The median and standard deviation (SD) were used to describe the quantitative variables. The normality distribution was verified using the Kolmogorov–Smirnov test with Lilliefors correction, and the homogeneity of variances was tested using the Levene test. Quantitative variables without a normal distribution were described using the median and interquartile range (25th–75th). Moreover, to analyze the effect of gender on the presence of MetS, a two-way ANOVA was performed, followed by Bonferroni's post hoc test. As regards the variables that violated the homogeneity of variations, the non-parametric Mann–Whitney U test was utilized to compare the effect of gender and the presence of MetS. Further, the receiver operating characteristic curves (ROC) were applied to identify the specificity and sensitivity of the independent variables' cut-off points in diagnosing MetS, along with positive and negative predictive values. The areas under the curve (AUC) and the 95% confidence intervals (CI) were also determined. The cut-off point was established by the CI of Youden's index. The significance of 5% (*p*-value $\leq$ 0.05) was adopted, and the analyses were performed using the SPSS version 24.0 for Windows, whereas the MedCalc version 15.8 s was used for the ROC analysis.

## 3. Results

Table 1 shows the study's included population's main anthropometrical and laboratory characteristics and information on eight IR indexes. These data were also compared for interaction between gender and the presence or absence of MetS. In all, 268 individuals were included in this study. Of these, 152 were male, and 116 were female. Of the male population, 70 were diagnosed with MetS during the conduction of this clinical study, and of the female population, 41 were diagnosed with MetS during the clinical study occurrence. Female individuals without MetS presented a higher mean age (59.8 $\pm$ 11.8). Groups with

MetS demonstrated 58 ± 14.7 (male) and 55.6 ± 14.1 (female) mean ages but without any significant differences. Male individuals with MetS presented lower levels of HDL-c (38.9 ± 9.7); interestingly, male subjects without MetS presented higher levels of HDL-c (54.0 ± 11.9). Regarding NC, there was no significant difference between the presence or absence of MetS and higher values of the variable, but there was a considerable difference between the genders ($p < 0.001$). Regarding WC, the difference was significant only while considering the MetS diagnosis ($p < 0.001$), as well as with HDL-c ($p < 0.001$), TG/FG ($p < 0.001$), WHtR ($p < 0.001$), TyG ($p < 0.001$), TyG-BMI ($p = 0.001$), TyG-WC ($p < 0.001$), TyG-WHtR ($p < 0.001$), TyG-NC ($p < 0.001$), TyG-NHtR ($p < 0.001$), and METS-IR ($p < 0.001$). The difference was significant while considering gender with WHtR ($p < 0.001$), NHtR ($p = 0.003$), TyG-WHtR ($p < 0.001$), and TyG-NC ($p < 0.001$).

**Table 1.** Comparison of mean and standard deviation (SD) for gender and metabolic syndrome (MetS).

| Variables | Men with MetS (*n* = 70) Mean | SD | Men without MetS (*n* = 82) Mean | SD | Women with MetS (*n* = 41) Mean | SD | Women without MetS (*n* = 75) Mean | SD | Anova *p*-Value Sex | MetS | Interaction |
|---|---|---|---|---|---|---|---|---|---|---|---|
| Age | 58.0 | 14.7 | 53.1 [†] | 13.9 | 55.6 | 14.1 | 59.8 [‡] | 11.8 | 0.214 | 0.853 | 0.009 *** |
| BMI (kg/m$^2$) | 29.5 | 4.8 | 30.0 | 5.9 | 30.7 | 7.8 | 29.3 | 5.2 | 0.740 | 0.539 | 0.201 |
| NC (cm) | 40.8 | 3.5 | 40.5 | 4.3 | 36.3 [‡] | 3.1 | 36.0 [‡] | 3.8 | <0.001 * | 0.469 | 0.941 |
| WC (cm) | 110.4 | 13.7 | 94.1 [†] | 13.9 | 106.0 | 11.7 | 97.1 [†] | 12.3 | 0.691 | <0.001 ** | 0.028 *** |
| HDL-c (mg/dL) | 38.9 | 9.7 | 54.0 [†] | 11.9 | 41.1 | 11.2 | 48.9 [†,‡] | 16.8 | 0.371 | <0.001 ** | 0.026 *** |
| TG/FG | 2.15 | 0.10 | 2.01 [†] | 0.13 | 2.18 | 0.10 | 2.01 [†] | 0.09 | 0.223 | <0.001 ** | 0.252 |
| WHtR | 0.64 | 0.08 | 0.54 [†] | 0.09 | 0.66 | 0.08 | 0.61 [†,‡] | 0.08 | <0.001 * | <0.001 ** | 0.044 *** |
| NHtR | 0.24 | 0.02 | 0.24 | 0.03 | 0.22 [‡] | 0.02 | 0.22 [‡] | 0.02 | 0.003 * | 0.716 | 0.818 |
| TyG | 9.19 | 0.46 | 8.57 [†] | 0.58 | 9.34 | 0.48 | 8.57 [†] | 0.45 | 0.223 | <0.001 ** | 0.252 |
| TyG-BMI | 271.1 | 46.7 | 257.6 | 58.9 | 285.8 | 68.7 | 250.9 [†] | 46.2 | 0.555 | 0.001 ** | 0.123 |
| TyG-WC | 1013.5 | 127.9 | 809.1 [†] | 141.5 | 991.3 | 125.9 | 833.5 [†] | 112.2 | 0.947 | <0.001 ** | 0.151 |
| TyG-WHtR | 5.89 | 0.78 | 4.71 [†] | 0.88 | 6.20 [‡] | 0.81 | 5.26 [†,‡] | 0.77 | <0.001 * | <0.001 ** | 0.250 |
| TyG-NC | 375.5 | 39.1 | 347.3 [†] | 47.9 | 339.6 [‡] | 33.8 | 308.9 [†,‡] | 33.8 | <0.001 * | <0.001 ** | 0.800 |
| TyG-NHtR | 2.18 | 0.23 | 2.02 [†] | 0.29 | 2.13 | 0.23 | 1.94 [†] | 0.21 | 0.043 * | <0.001 ** | 0.752 |
| METS-IR | 49.3 | 9.7 | 43.5 [†] | 9.9 | 51.0 | 14.0 | 43.9 [†] | 9.0 | 0.435 | <0.001 ** | 0.614 |

* indicates the main effect of a group's independent sex (MetS) by a two-way ANOVA test for *p*-value < 0.05; ** indicates group main effect (MetS) regardless of sex by ANOVA-two-way test for *p*-value < 0.05; *** indicates a significant interaction effect between group and sex by ANOVA-two-way test for *p*-value < 0.05; [†] suggests a significant difference concerning the group with MetS within each sex by the Post-hoc Bonferroni test for *p*-value < 0.05; [‡] indicates a significant difference concerning males within each group (MetS) by the Post-hoc Bonferroni test for *p*-value < 0.05.

Quantitative variables without a normal distribution were described using the median and interquartile range (25th–75th). For the variables that violated the assumption of variances' homogeneity, it was also necessary to perform the non-parametric Mann–Whitney test to compare sex and MetS. These variances and their corresponding comparisons are presented in Table 2. Assuming that hypertension and the state of IR are the most frequent parameters in individuals with MetS, men with MetS showed the highest levels of FG (122.88 ± 38.77) compared with all other groups. However, women diagnosed with MetS presented the highest levels of TG (224.7 ± 100.9) than the other groups, and this difference was significant concerning the male group. Men with MetS also gave the most increased

SBP and DBP values, 136.90 ± 16.34 and 86.09 ± 10.28, respectively. In turn, women with MetS presented the highest TG-HDL-c (6.11 ± 4.31) values.

**Table 2.** Comparison of the median and interquartile range (25th–75th) for variables without normal distribution.

| | Sex | | | | | | | | | | | |
| --- | --- | --- | --- | --- | --- | --- | --- | --- | --- | --- | --- | --- |
| | **Men** | | | | | | **Women** | | | | | |
| **Variables** | **MetS** | | | | | | | | | | | |
| | **With MetS (*n* = 70)** | | | **Without MetS (*n* = 82)** | | | **With MetS (*n* = 41)** | | | **Without MetS (*n* = 75)** | | |
| | **Median** | **25th** | **75th** | **Median** | **25th** | **75th** | **Median** | **25th** | **75th** | **Median** | **25th** | **75th** |
| Fasting glucose (mg/dL) | 109.0 | 98.7 | 12.8.5 | 94.0 † | 88.0 | 99.2 | 106.0 | 95.0 | 125.0 | 93.8 † | 88.2 | 101.0 |
| TG (mg/dL) | 161.0 | 126.5 | 222.5 | 106.4 † | 75.6 | 150.7 | 187.0 ‡ | 165.0 | 274.0 | 108.0 † | 83.0 | 158.0 |
| Systolic blood pressure (mmHg) | 140.0 | 120.0 | 150.0 | 120.0 † | 120.0 | 130.0 | 137.0 | 120.0 | 155.0 | 120.0 † | 120.0 | 130.0 |
| Diastolic blood pressure (mmHg) | 80.0 | 80.0 | 100.0 | 80.0 † | 80.0 | 80.0 | 80.0 | 80.0 | 100.0 | 80.0 † | 80.0 | 80.0 |
| TG-HDL-c | 4.11 | 3.07 | 5.94 | 2.04 † | 1.40 | 3.19 | 4.81 | 3.63 | 7.54 | 2.38 † | 1.69 | 3.69 |

† indicates a significant difference concerning the group with MetS within each sex by the non-parametric Mann–Whitney test for *p*-value < 0.05; ‡ indicates a significant difference concerning males within each group (MetS) by the non-parametric Mann–Whitney test for *p*-value < 0.05.

The results of ROC and AUC analysis, the 95% confidence interval, optimal thresholds, corresponding sensitivity, corresponding specificity, and Youden's index values for the whole study group by each IR index are presented in Table 3. The ROC of each IR index is shown in Figure 1. According to Table 3, all IR indexes acquired significant accuracy in diagnosing MetS in the studied population. These were >8.882048782 for TyG, >249.3913555 for TyG-BMI, >860.7463699 for TyG-WC, >5.365297405 for TyG-WHtR, >328.0513282 for TyG-NC, >1.9845651 for TyG-NHtR, >2.552631579 for TG-HDL-c, and >43.82124867 for METS-IR. The most significant AUC were for TyG-WC (0.849 (0.800–0.889)), TyG (0.837 (0.787–0.879)), and TG-HDL-c (0.817 (0.765–0.861)). The lowest AUC were for TyG-BMI (0.630 (0.569–0.688)), METS-IR (0.683 (0.623–0.738)), and TyG-NHtR (0.713 (0.654–0.766)). The most remarkable sensitivities were for TG-HDL-c (90.99%), TyG-WC (89.19%), and TyG-NC (84.68%), and the lowest were for METS-IR (68.47%), TyG-BMI (70.27%), and TyG-WHtR (79.28%). The greatest specificities were for TyG (73.89%), TyG-WHtR (72.61%), and TyG-WC (66.88%), and the lowest were for TyG-BMI (53.5%), TyG-NHtR (56.05%), and TyG-NC (56.69%).

**Table 3.** Analysis of the area under the curve (AUC), sensitivity, specificity, positive predictive value (+PV), and negative predictive value (−PV) with their respective 95% confidence intervals (95% CI) of the cut-off points for the variables independent for the diagnosis of metabolic syndrome (MetS).

| Variables | Cut-off Points for MetS | AUC (IC95%) | Sensitivity | 95% CI | Specificity | 95% CI | +PV | 95% CI | −PV | 95% CI |
| --- | --- | --- | --- | --- | --- | --- | --- | --- | --- | --- |
| WHtR | >0.607361963 | 0.715 (0.657–0.768) * | 70.27 | 60.9–78.6 | 66.24 | 58.3–73.6 | 59.5 | 50.6–68.0 | 75.9 | 67.9–82.8 |
| NHtR | >0.219298246 | 0.541 (0.479–0.601) | 76.58 | 67.6–84.1 | 36.31 | 28.8–44.3 | 45.9 | 38.6–53.4 | 68.7 | 57.6–78.4 |
| TyG | >8.882048782 | 0.837 (0.787–0.879) * | 83.78 | 75.6–90.1 | 73.89 | 66.3–80.6 | 69.4 | 60.9–77.1 | 86.6 | 79.6–91.8 |
| TyG-BMI | >249.3913555 | 0.630 (0.569–0.688) * | 70.27 | 60.9–78.6 | 53.5 | 45.4–61.5 | 51.7 | 43.4–59.9 | 71.8 | 62.7–79.7 |
| TyG-WC | >860.7463699 | 0.849 (0.800–0.889) * | 89.19 | 81.9–94.3 | 66.88 | 58.9–74.2 | 65.6 | 57.4–73.1 | 89.7 | 82.8–94.6 |

**Table 3.** *Cont.*

| Variables | Cut-off Points for MetS | AUC (IC95%) | Sensitivity | 95% CI | Specificity | 95% CI | +PV | 95% CI | −PV | 95% CI |
|---|---|---|---|---|---|---|---|---|---|---|
| TyG-WHtR | >5.365297405 | 0.804 (0.751–0.850) * | 79.28 | 70.5–86.4 | 72.61 | 64.9–79.4 | 67.2 | 58.4–75.1 | 83.2 | 75.9–89.0 |
| TyG-NC | >328.0513282 | 0.722 (0.664–0.774) * | 84.68 | 76.6–90.8 | 56.69 | 48.6–64.6 | 58 | 50.0–65.7 | 84 | 75.6–90.4 |
| TyG-NHtR | >1.9845651 | 0.713 (0.654–0.766) * | 82.88 | 74.6–89.4 | 56.05 | 47.9–64.0 | 57.1 | 49.1–64.9 | 82.2 | 73.7–89.0 |
| TG-HDL-c | >2.552631579 | 0.817 (0.765–0.861) * | 90.99 | 84.1–95.6 | 63.06 | 55.0–70.6 | 63.5 | 55.5–71.0 | 90.8 | 83.8–95.5 |
| METS-IR | >43.82124867 | 0.683 (0.623–0.738) * | 68.47 | 59.0–77.0 | 59.87 | 51.8–67.6 | 54.7 | 46.0–63.1 | 72.9 | 64.3–80.3 |

* indicates a significant effect of the cut-off point for the diagnosis of MetS for *p*-value < 0.0001 (the confidence interval of the Youden index established the cut-off point.

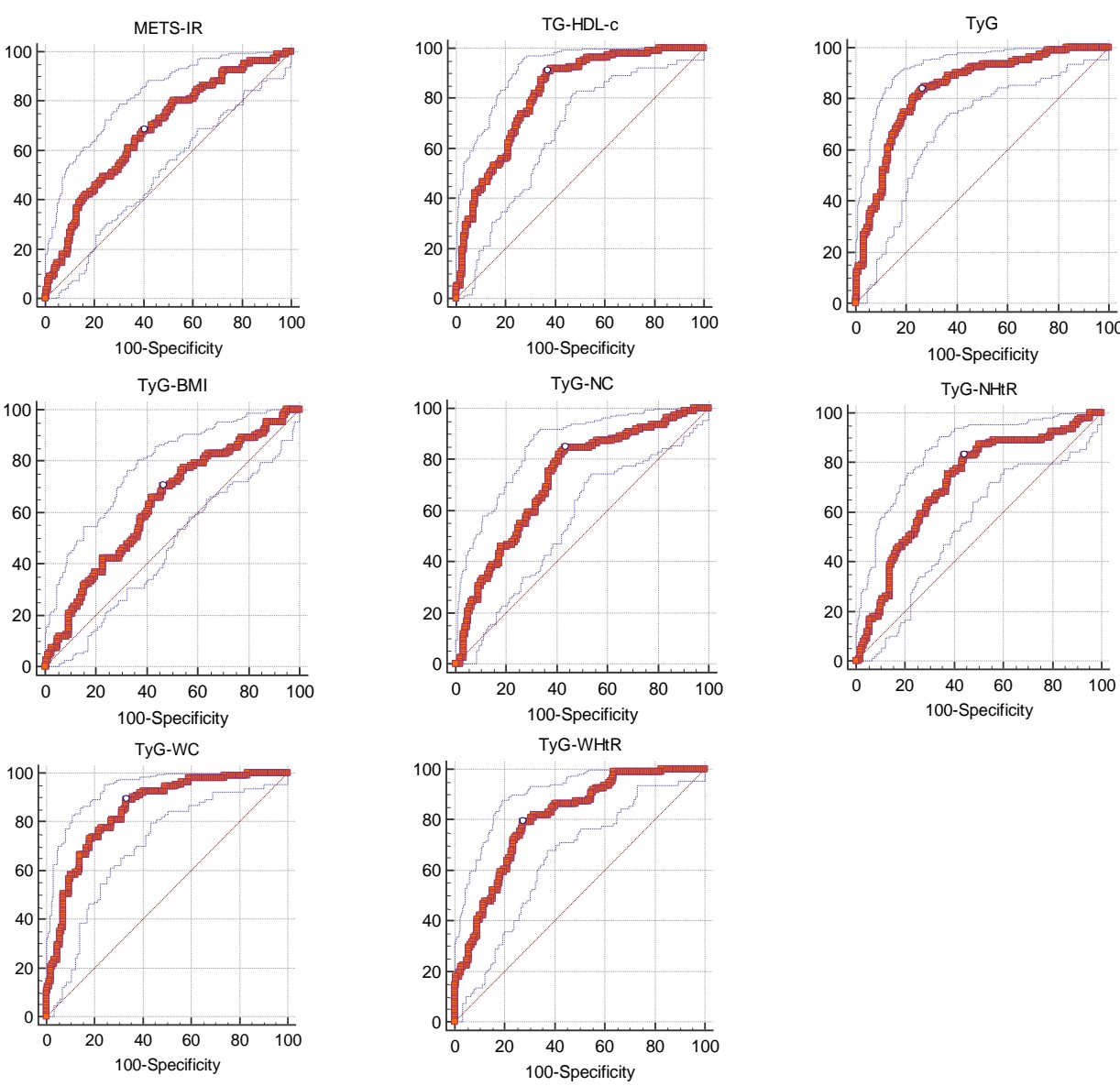

**Figure 1.** Analysis of the area under the curve (AUC), sensitivity, and specificity of cut-off points for independent variables for diagnosing metabolic syndrome (MetS).

## 4. Discussion

This study has found that TyG-WC, TyG, and TyG-WHtR reach the greatest AUC values among all the indexes, which suggests that they are the most useful diagnostic indicators of MetS for individually directing the clinical management of Brazilian patients with MetS. However, all indexes were found to achieve significant accuracy, with the highest AUC values for TyG-WC (0.849), TyG (0.837), and TyG-WHtR (0.804). Additionally, the most heightened diagnostic sensitivities were observed for TyG-WC (89.19%), TyG-NC (84.68%), and TyG (83.78%). These findings are consistent with those found in the existing literature.

Huang et al. [24] conducted a clinical study on 569 middle-aged Chinese individuals to assess the usefulness of four surrogate IR indexes in the diagnosis of IR. This study on a population of 67.7% men with a mean age of 48.5 demonstrated that in comparison to the visceral adiposity index (VAI) and TG/HDL-C, the lipid accumulation product (LAP) and TyG have a higher predictive ability to diagnose IR in the Chinese population. Further, compared with other IR indicators, HOMA-IR demonstrated significant positive correlations with TG-HDL-c, VAI, LAP, and TyG, i.e., 0.306, 0.217, 0.381, and 0.371, respectively. Among the four IR indicators studied, LAP presented the highest specificity and TyG the most heightened sensitivity. The AUC value to predict IR diagnosed previously with HOMA-IR was 0.773 for TG-HDL-c, 0.806 for LAP, 0.767 for VAI, and 0.800 for TyG.

Demir et al. [25] also studied the homeostasis model assessment-estimated insulin resistance 1 (HOMA1-IR) and 2 (HOMA2-IR) to evaluate their optimal threshold values in diagnosing IR among Turkish individuals. They identified the cut-off points in which these indexes exerted a higher diagnostic accuracy, which was 2.46 in HOMA1-IR and 1.40 in HOMA2-IR. Further, by using the HOMA2-IR method, they found that the overall prevalence of IR among the participants in 2013 was 33.2%, being higher among women (35.6%) than men (30.1%).

IR indicators have been investigated in different populations across the world to determine their effectiveness in predicting IR and its related complications, such as CVDs. For instance, Er et al. [26] utilized IR indexes to assess IR in non-diabetic Taiwanese individuals. A total of 511 patients were enrolled for the final analyses, and the clinical usefulness of various parameters was analyzed. It was found that for all used lipid-derived IR indicators, TG-HDL-c exhibited higher additional variations compared to HOMA-IR (7% in total). Although TG-HDL-c presented a higher AUC value (0.707) in ROC compared to other lipid-derived IR indicators, TyG-BMI had the highest AUC value (0.801) of all indicators used in this clinical study. Even so, only TyG-BMI demonstrated any significant benefit in predicting IR in this cohort of patients due to its robust association with HOMA-IR's predictor power. The results of the present study elucidate that TyG-WC was one of the most useful indexes in predicting MetS, reaching an AUC value of 0.772.

In their study, García et al. [27] reported a high frequency of low HDL-c and hypertriglyceridemia among Mexican children and postulated that TyG and TG-HDL-c could function as triage predictors of IR in this population. Overweight children between the ages of five and nine participated in the clinical study. In total, 104 normal-weight and 97 overweight children were included as participants. The results revealed that TyG had a cut-off point of 8.5 and an AUC value of 0.802 (with IC95% = 0.77–0.893 and diagnostic accuracy = 73%), demonstrating better diagnostic accuracy than TG-HDL-c, which presented a cut-off point of 2.22 and an AUC value of 0.729 (with an IC95% = 0.622–0.837 and diagnostic accuracy = 52%). Other novel risk factors such as perinatal factors, nutrigenomics, diet, nutri-epigenetics, hyperuricemia, cardiorespiratory fitness, and dyslipidemia are associated with the link between childhood obesity as well as IR and the occurrence of CVDs in adulthood [28]. Therefore, it is necessary to identify the alterations in the onset of adult life to delay the occurrence of chronic diseases related to childhood obesity, such as hypertension and other CVDs.

Aslan Çin et al. [29] aimed to determine the cut-off points of TG-HDL-c, TyG, and HOMA-IR for diagnosing MetS among obese adolescents in Turkey. The participants

consisted of 1171 obese adolescents (532 men and 639 women) aged 10–16. The cut-off points for diagnosing MetS using TG-HDL-c, TyG, and HOMA-IR were found to be 2.16 (88.8% of sensitivity and 49.7% of specificity), 8.5 (85.6% of sensitivity and 57.0% of specificity), and 2.52 (83.2% of sensitivity and 40.4% of specificity), respectively. TyG and TG-HDL-c were found to be better markers than HOMA-IR for the diagnosis of MetS. This study was conducted according to the guidelines of the International Diabetes Federation (IDF). Mirr et al. [15] investigated TyG-NHtR and TyG-NC principally to assess their usefulness in diagnosing MetS in Poland. The participants in their study included 665 non-diabetic adult patients. They found that the two indexes presented high diagnostic accuracy and reached significant AUC values (i.e., 0.831 for TyG-NHtR and 0.791 for TyG-NC, with IC95% = 0.818–0.876 and 0.757–0.825, respectively).

To investigate the relationships between IR indexes and the risk of developing type 2 diabetes combined with hypertension, Dong et al. [30] conducted a study comprising 8892 participants, who were divided into two cohorts of 4234 and 4658. In multivariable-adjusted models, TyG was the indicator that demonstrated a higher risk of developing type 2 diabetes combined with hypertension, with an IC95% of 3.46 (2.43–4.93) and 2.02 (1.67–2.44), respectively. Additionally, the authors also found differences in the associations between TG-HDL-c, with T2DM and hypertension in the two cohorts.

Besides helping diagnose MetS, IR indexes can be further associated with other cardiometabolic consequences, such as subclinical atherosclerosis and arterial stiffness [31–33]. Chen et al. [34] aimed to investigate the associations between TyG and atrial fibrillation (AF) in a cohort of 356 hospitalized patients from China. In this retrospective observational study, the authors found that TyG became significantly higher in the AF group compared to the group without AF. Moreover, multivariate logistic regression revealed that TyG was positively associated with not only AF (IC95% = 1.412–3.100) but also hypertension (IC95% = 1.135–2.717). For TyG, the AUC value in associating with AF was 0.600 (IC95% = 0.542–0.659). The optimal critical value was determined to be 8.35, which demonstrated a higher sensitivity of 65.4% and specificity of 52.0%. These values correlated with higher TyG levels and confirmed it as an independent risk factor for AF among Chinese patients.

Wang et al. [35] enrolled 1576 participants who were not previously diagnosed with CVD to undergo multidetector computed tomography to screen for coronary artery calcification (CAC). After the examinations, the authors found that the increases in METS-IR values were independently associated with a higher prevalence of CAC. According to the AUC or ROC analyses, METS-IR's cut-off point at which this IR indicator predicted CAC was found to be 0.607.

Wu et al. [36] aimed to examine the relationship between TG-HDL-c, TyG, and METS-IR and the diagnosis of coronary artery disease (CAD). In total, 802 patients were included in the study and underwent coronary angiography. The results demonstrated that the TG-HDL-c, TyG, and METS-IR ratios increased in CAD patients and that TG-HDL-c and METS-IR were independently associated with the presence of CAD, with IC95% = 1.32 (1.02–1.70) and 1.65 (1.32–2.05), respectively. Conversely, METS-IR was an independent predictor of CAD severity, with IC95% = 1.22 (1.02–1.47).

The main objective of the present clinical study was to find out a simple and accurate index or group of indexes to be used clinically for predicting MetS in the at-risk Brazilian population. This is because the early detection of MetS is essential to facilitate the diagnostic process of this highly prevalent metabolic disorder as well as the clinical evaluation of other cardiometabolic consequences, such as subclinical atherosclerosis and arterial stiffness.

## 5. Strengths and Limitations

To the best of our knowledge, this is the first study to assess the role of the newly proposed IR indicators in detecting MetS among the Brazilian population. However, some limitations of the study must also be addressed. First, the study focused only on the Brazilian population; therefore, caution needs to be exercised when extrapolating these

findings to other ethnicities. Second, patients who take medications for hyperlipidemia, impaired glucose tolerance, or diabetes were excluded to create a homogenous group for the investigation of early indicators of MetS. Therefore, the proposed indicators should be interpreted as early markers of MetS in untreated patients. Further studies are required to determine which indexes are useful for diagnosing MetS among patients undergoing treatment for hyperlipidemia, impaired glucose tolerance, or diabetes.

## 6. Conclusions

Indirect IR indexes can diagnose not only MetS but also other cardiometabolic conditions that predispose the population to an increased risk of sudden death. In our observation, TyG-WC, TyG, and TyG-WHtR reached the greatest AUC values among all the indexes, which suggests that they are the most useful diagnostic indicators of MetS in individually guiding the clinical management of Brazilian patients with MetS.

**Author Contributions:** Conceptualization, L.F.L., G.M., R.J.T., K.Q., E.F.B.C. and S.M.B.; methodology, L.F.L., G.M., R.J.T., K.Q., E.F.B.C. and S.M.B.; software, L.F.L., G.M., R.J.T., K.Q., E.F.B.C. and S.M.B.; validation, L.F.L., G.M., R.J.T., K.Q., E.F.B.C. and S.M.B.; formal analysis, L.F.L., G.M., R.J.T., K.Q., E.F.B.C. and S.M.B.; investigation, L.F.L., G.M., R.J.T., K.Q., E.F.B.C. and S.M.B.; resources, L.F.L., G.M., R.J.T., K.Q., E.F.B.C. and S.M.B.; data curation, L.F.L., G.M., R.J.T., K.Q., E.F.B.C. and S.M.B.; writing—original draft preparation, L.F.L., G.M., R.J.T., K.Q., E.F.B.C. and S.M.B.; writing—review and editing, L.F.L., G.M., R.J.T., K.Q., E.F.B.C. and S.M.B.; visualization, L.F.L., G.M., R.J.T., K.Q., E.F.B.C. and S.M.B.; supervision, L.F.L., G.M., R.J.T., K.Q., E.F.B.C. and S.M.B.; project administration, L.F.L., G.M., R.J.T., K.Q., E.F.B.C. and S.M.B.; funding acquisition, L.F.L., G.M., R.J.T., K.Q., E.F.B.C. and S.M.B. All authors have read and agreed to the published version of the manuscript.

**Funding:** This research received no external funding.

**Institutional Review Board Statement:** The study was conducted in accordance with the Declaration of Helsinki, and approved by the Institutional Review Board (Ethics Committee of the University of Marília) under the ethical approval number 50817221.6.0000.5496 on 30 September.

**Informed Consent Statement:** Informed consent was obtained from all subjects involved in the study.

**Data Availability Statement:** Not applicable.

**Acknowledgments:** The authors acknowledge the Núcleo Integrado de Pesquisa e Extensão (NIPEX) of University of Marília for supporting the conduction of this research.

**Conflicts of Interest:** The authors declare no conflict of interest.

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
