# Peer review of "Detection of Metabolic Syndrome Using Insulin Resistance Indexes: A Cross-Sectional Observational Cohort Study"

_endocrines, doi:10.3390/endocrines4020021_

Round 1

Reviewer 1 Report

General points: 

Metabolic syndrome is a constellation of high risk diseases as evidenced by IR, and has to meet 3/5 features. I have never used these 'diagnostic indicators' in my clinical practice. can you tell me what is the purpose of these 'diagnostic indicators' when the vitals and labs can tell us if a person as Metabolic syndrome or not. MetS doesnt cause diseases, but is only a marker of high risk as it is a constellation of diseases which are dangerous. Just like AIDS doesnt cause infection but HIV does. 

If patient has high triglycerides for reason other than insulin resistance, that would change the entire indices since TG's are dependant on fasting vs non fasting state. 

Please use the words 'Indexes' and 'indices' appropriately in your paper. there have been used interchangably in some places where it should not happen. 

Please have this reviewed by native english speaker to improve the strength of medical vocabulary. 

Introduction: 

1. Mets is not just due to lifestyle changes, but they are multifactorial. if you are introducing MetS, please given details about its multifactorial nature. 

2. Please improve the Introduction. the 1st 2 sentences do not make much sense and do not help your topic in anyway. 

3. the word pro-inflammatory has been repeated twice in1 sentences. 

4. the gold standard method to 'quantify insulin sensitvity' 

5. Block of the pancreas--> reduction in pancreatic and hepatic glucose production by increased glucose disposal into tissues. 

Materials and Methods:

2.2--> please write chronologically and clearly--> inclusion criteria and exclusion criteria and dont mix them up. 

2.3--> you said this was an observational study. How did you measure NC and WC for your patients? do you that with all your patients in cardiology clinic. The study was approved in September 2021 but you included patients from June 2021 to December 2021. was this an Observational or prospective study ?  Instead of writing WHtR is defined as----> you can say waist to height Ratio[WHtR] is calculated as the .......  Its an acronym not a definition. do it for NHtR as well. 

2.6--> is there any worldwide difference if metabolic syndrome criteria by ethnicity ? is it different for hispanics and asians ? their waist size is lesser it seems. do you want to mention that. 

2.8--> in the TG-glucose index, what is this Ln ? 

Discussion:

In demir et al writeup--> 'the results showed the cut-off popints in which these indexes exerted higher diagnostic accuracy'- please expand on this. 

In Er et al. --> you wrote 'authors from al over the world'- I looked up and all authors are from taiwan. please fic that entire paragraph.  at the end of the paragraph, you mentioned 'in comparison with our results'--> and you mentioned results of your study. did you want to mention your results or their results in ER et al writeup ? 

Garcia et al. --> 'studies like this are crucial.....- doesnt make sense and not enough teaching point. please improve those sentences'. 

You mentioned huang et al-reference 15, before DEMIR et al, and again after garcia et al. this has been repeated. Should have double checked your content before submitting. 

'in turn, to predeict and diagnose MetS with use of IR indexes also, a few studies were published.'--> this sentence doesnt add anything to this paragraph. you can avoid this. 

you wrote about dong et al and IR indexes and risk of developing DM2, but I thought we are talking about metabolic syndrome and not DM2. 

"IR indexes can be further associated with other Cardiometabolic consequences, such as subclinical atherosclerosis and arterial stiffness'- where is the reference. MetS causes much more than these above stated diseases. 

the last paragraph of your discussion is very long. needs to be broken into 2 or 3 sentences. 

Paragraph of an article should not have more than 6-7 sentences. some of your paragraphs are too long. 

I see that all your discussion is about other studies who included such similar indexes of IR. I personally do not learn much from your study[not sure if obs or prospective]. It would be much wiser to do this as a systematic review or meta analysis, since you have quoted so many studies. A Table of all IR related indices and various studies would look really well. 

I do not see any references to Dr. Grundy [person who coined metabolic syndrome] or Dr Reaven [who coined syndrome X].  https://pubmed.ncbi.nlm.nih.gov/16157765/

Author Response

The authors of this manuscript express their sincere thanks to the Assistant Editor and reviewers for critically assessing this work. The authors have acted upon the recommendations of the editors and reviewers, which have significantly enhanced the quality of this manuscript. All modifications incorporated in the manuscript are highlighted in the red color font. A "point-by-point" response to each comment is outlined below.

Comment 1:

Metabolic syndrome is a constellation of high-risk diseases, as evidenced by IR, and must meet 3/5 features. I have never used these 'diagnostic indicators' in my clinical practice. Can you tell me the purpose of these 'diagnostic indicators' when the vitals and labs can say to us if a person has Metabolic syndrome? MetS doesn't cause diseases, but it is only a marker of high risk as it is a constellation of dangerous conditions. Just like AIDS doesn't cause infection, HIV does. 

Response:

            We sincerely appreciate this comment. As insulin resistance can be considered cardinal to the pathophysiology of metabolic syndrome, clinical indicators that assess glucose intolerance also become more manageable and more uncomplicated in diagnosing metabolic syndrome. These indexes diagnose if a person is or is not insulin resistant. Suppose diagnosing a person with metabolic syndrome is possible based on simple equations and not on demanding diagnostic criteria. In that case, the clinical care of patients at cardiovascular risk is facilitated.

Comment 2:

If a patient has high triglycerides for reasons other than insulin resistance, that will change the entire indices since TG's depend on fasting vs. non-fasting state. 

Response:

            In our study, the laboratory measurements followed the instructions of the University Hospital's laboratory, and all TG measurements were collected upon patient fasting. We added this information to our main document. Please,  see line 137.

Comment 3:

Please appropriately use the words ‘'Indexes'’ and '’Indices’' in your paper. There have been used interchangeably in some places where it should not happen. 

Response:

            We made the necessary revisions throughout our manuscript. We searched for incorrect terms such as "indices," as mentioned by you, and all the words related to IR equations were standardized to "Index" or "Indicator" as the primary manuscripts that were previously published about this issue.

Comment 4:

Please have this reviewed by a native English speaker to improve the strength of your medical vocabulary. 

Response:

            We appreciate this suggestion. We hired PaperTrue to proofread our manuscript.

Introduction: 

Comment 5:

MetS is not just due to lifestyle changes, but they are multifactorial. If you are introducing MetS, please give details about its multifactorial nature. 

Response:

            You are correct that MetS is multifactorial, and we must have better elucidated this idea in the manuscript. We added the required details in lines 41, 46-48, 49, and 50-53. We also added sentences about atherosclerosis in lines 54-60 and supplemented the concept of oxidative stress in lines 48-51.

Comment 6:

Please improve the Introduction. The 1st two sentences do not make much sense or help your topic. 

Response:

            We removed the 1st two sentences of the Introduction. Please, see lines 1-2.

Comment 7:

The word pro-inflammatory is repeated twice in 1 sentence. 

Response:

            We removed the duplicates. Please, see lines 46-47.

Comment 8:

The standard gold method to 'quantify insulin sensitivity.' 

Response:

            We changed the word. Please, see line 68.

Comment 9:

Block of the pancreas--> reduction in pancreatic and hepatic glucose production by increased glucose disposal into tissues. 

Response:

            Thank you so much for this suggestion. We added this sentence in the manuscript. Please see lines 72-73.

Materials and Methods:

Comment 10:

2.2--> please write chronologically and clearly--> inclusion and exclusion criteria and don't mix them up. 

Response:

            We appreciate this suggestion. Please see lines 103-108 for the required corrections.

Comment 11:

2.3--> you said this was an observational study. How did you measure NC and WC for your patients? Do you do that with all your patients in the cardiology clinic? The study was approved in September 2021, but you included patients from June 2021 to December 2021. was this an Observational or prospective study?  Instead of writing WHtR is defined as---->, you can say waist to height Ratio[WHtR] is calculated as the .......  It's an acronym, not a definition. Could you do it for NHtR as well? 

Response:

            Thank you for this comment. 1) We reclassified our study as a cross-sectional observational cohort (please, see lines 3, 90, and 128) as we used medical records to conduct the investigations. Despite this alteration, the methodology remains unchanged. 2) There was a mistake in the month of the approval. We corrected it in the manuscript. Please see line 169-170. 3) We used your recommendations for WHtR and NHtR. Please see lines 125-127.

Comment 12:

2.6--> is there any worldwide difference if metabolic syndrome criteria by ethnicity? Is it different for Hispanics and Asians ? their waist size is lesser, it seems. Do you want to mention that?

Response:

            Thank you for this comment. We added this information on lines 147-150.

Comment 13:

2.8--> in the TG-glucose index, what is this Ln?

Response:

            Ln is a natural logarithm.

Discussion:

Comment 14:

In Demir et al. writeup--> 'the results showed the cut-off points in which these indexes exerted higher diagnostic accuracy'- please expand on this. 

Response:

            We expanded the paragraph with the authors' information, Please, see lines 74-78 of the discussion.

Comment 15:

In Er et al. --> you wrote 'authors from all over the world'- I looked up, and all authors are from Taiwan. Please fix that entire paragraph.  At the end of the section, you mentioned 'in comparison with our results'-->, and you said the results of your study. Did you want to mention your or their results in ER et al. write-up? 

Response:

            1) We intended to register that authors worldwide worked with IR indexes to predict MetS in their respective populations. We rewrote the sentence maintaining our initial idea. Please, see lines 79-80 of the discussion. 2) We intended to compare the Taiwanese results with ours to reiterate the population-specificity of IR indexes. We rewrote the sentence to make it clear. Please, see lines 89-90 of the discussion.

Comment 16:

Garcia et al. --> 'studies like this are crucial.....- doesn't make sense and not enough teaching point. Please improve those sentences'. 

Response:

            We improved these sentences. Please, see lines 102-104 of the discussion.

Comment 17:

You mentioned Huang et al.-reference 15, before DEMIR et al., and again after Garcia et al. this has been repeated. You should have double-checked your content before submitting it. 

Response:

            We reorganized the text. Please, see lines 62-71 and 72-78 of the discussion.

Comment 18:

'in turn, to predict and diagnose MetS with the use of IR indexes also, a few studies were published.'--> this sentence does not add anything to this paragraph. You can avoid this. 

Response:

            We excluded this sentence.

Comment 19:

You wrote about Dong et al. and IR indexes and the risk of developing DM2, but I thought we were talking about metabolic syndrome and not DM2. 

Response:

Thank you so much for this concern. As you know, IR is cardinal for MetS and is also the pathway for DM2. For these reasons, we added Dong et al.'s study to our article.

Comment 20:

"IR indexes can be further associated with other Cardiometabolic consequences, such as subclinical atherosclerosis and arterial stiffness'- where is the reference? MetS causes much more than these above-stated diseases. 

Response:

            We added three new references. Please, see in the references list the numbers 31 (lines 268-270), 32 (lines 271-274), and 33 (lines 275-277).

Comment 21:

The last paragraph of your discussion is very long. It needs to be broken into 2 or 3 sentences. Paragraphs of an article should not have more than 6-7 sentences. Some of your paragraphs are too long. 

Response:

Thank you so much for this concern. We broked the last discussion paragraph. Please, see discussion section’s lines 127-136 for the 1st paragraph, 137-142 for the 2nd paragraph, and 143-149 for the 3rd paragraph.

Comment 22:

I see that your discussion is about other studies that included similar IR indexes. I do not learn much from your research [unsure if obs or prospective]. Doing this as a systematic review or meta-analysis would be much wiser since you have quoted many studies. A Table of all IR-related indices and various studies would look well. 

Response:

            Dear reviewer, we agree with you and we can program to systematically review the literature to meta-analyze the current IR-related indexes data. However, our study is the first in our knowledge throughout the scientific literature to evaluate the cut-off points of the newly proposed IR indexes among the Brazilian population in diagnosing MetS. So, our study is a significant contribution as IR indexes cut-off diagnostic points are population-dependent; therefore, studies are necessary for each population.

Comment 23:

I do not see references to Dr. Grundy [person who coined metabolic syndrome] or Dr. Reaven [who coined syndrome X].  https://pubmed.ncbi.nlm.nih.gov/16157765/

Response:

Thank you for this reminder. We cited these authors in references number 4 and 5. Please see lines 202-206 of the references list.

Dear reviewer, thank you so much for revising our manuscript. We know your time is precious, but your suggestions improved our work's quality! We are grateful!

Reviewer 2 Report

I have received for review an original research article entitled “Insulin Resistance Indexes Detecting Metabolic Syndrome: A Cross-Sectional Study” prepared by Lucas Fornari Laurindo et al., which is being processed for publication in the journal Endocrines. Insulin resistance is the basic pathogenetic mechanism present in type 2 diabetes. Type 2 diabetes is a growing problem in the modern world due to epidemic of the obesity and the metabolic syndrome. Diabetes is a strong factor leading to the development of cardiovascular diseases, which in turn are the leading cause of morbidity and mortality worldwide. Therefore, conducting research on such topics is extremely important. The efforts of the authors who have undertaken such an important research topic should be appreciated. The submitted manuscript has quite a high scientific and cognitive value and should be considered for publication in the future. However, some significant improvements are needed. My suggestions are listed below.

1)     In my opinion, the introduction needs to be supplemented. It is worth mentioning that cardiovascular diseases in the course of atherosclerosis are the main cause of morbidity and mortality in the world, and type 2 diabetes and metabolic syndrome are strong risk factors for their development. The most important atherosclerotic cardiovascular diseases include ischemic heart disease, cerebrovascular disease and peripheral arterial disease. Angioplasty and stenting play an important role in the treatment of atherosclerotic cardiovascular disease, but the phenomenon of restenosis significantly reduces their effectiveness and may lead to the need for re-intervention. (10.3390/ijerph182211970) When discussing the topic of oxidative stress in the context of metabolic syndrome, it is worth mentioning that according to current research, the obesity and insulin resistance is the component of the metabolic syndrome that contributes most to the relationship between the metabolic syndrome and oxidative stress. (10.3390/antiox11010079) It is also worth mentioning that oxidative modification of lipoproteins may make them more atherogenic. An example of such modified lipoproteins are nitrated lipoproteins, the importance of which in the pathogenesis of cardiovascular disease in diabetic patients has recently been discussed.

2)     In describing the methodology (2.2.), the authors used the term: "other drugs that could interfere with the metabolism". Be very careful with that term. Many drugs, in addition to their main mechanism of action, have additional mechanisms, often not yet fully understood. Please explain exactly what drugs the Authors understand by this term.

3)     The study was conducted from June to December, and the positive opinion of the bioethics committee was obtained in September of the same year. Please explain.

4)     The description of the statistical analysis methodology needs improvement. Describe how the compliance of the distribution of a given variable with the normal distribution was tested. In the case of variables whose distribution is consistent with the normal distribution, the adequate measure of central tendency and dispersion is the arithmetic mean and standard deviation, and in the case of variables whose distribution differs significantly from the normal distribution, the median and interquartile range are adequate parameters.

5)     Sudden cardiac death, arterial stiffness, and subclinical atherosclerosis were not observed and examined according to the methodology and results presented, so I do not understand why such topics are brought up in the conclusions and at the end of the discussion.

6)     Strengths and limitations of the study should be discussed.

7)     The number of references should be increased (currently it is very small).

8)     Although English is without huge mistakes and it is understandable, it would be worth to perform the language correction by the specialist.

Author Response

REVIEWER 2

The authors of this manuscript express their sincere thanks to the Assistant Editor and reviewers for critically assessing this work. The authors have acted upon the recommendations of the editors and reviewers, which have significantly enhanced the quality of this manuscript. All modifications incorporated in the manuscript are highlighted in the red color font. A "point-by-point" response to each comment is outlined below.

Comment 1:

I have received for review an original research article entitled "Insulin Resistance Indexes Detecting Metabolic Syndrome: A Cross-Sectional Study," prepared by Lucas Fornari Laurindo et al., which is being processed for publication in the journal Endocrines. Insulin resistance is the basic pathogenetic mechanism present in type 2 diabetes. Type 2 diabetes is a growing problem in the modern world due to the epidemic of obesity and metabolic syndrome. Diabetes is a decisive factor leading to cardiovascular disease development, the leading cause of morbidity and mortality worldwide. Therefore, researching such topics is extremely important. The authors' efforts have undertaken such an important research topic should be appreciated. The submitted manuscript has quite a high scientific and cognitive value and should be considered for publication in the future. However, some significant improvements are needed. My suggestions are listed below.

Comment 2:

In my opinion, the Introduction needs to be supplemented. It is worth mentioning that cardiovascular diseases in atherosclerosis are the leading cause of morbidity and mortality worldwide, and type 2 diabetes and metabolic syndrome are decisive risk factors for their development. The essential atherosclerotic cardiovascular diseases include ischemic heart disease, cerebrovascular disease, and peripheral arterial disease. Angioplasty and stenting play a crucial role in treating atherosclerotic cardiovascular disease. Still, the phenomenon of restenosis significantly reduces their effectiveness and may lead to the need for re-intervention. (10.3390/ijerph182211970). When discussing the topic of oxidative stress in the context of metabolic syndrome, it is worth mentioning that according to current research, obesity, and insulin resistance is the component of the metabolic syndrome that contributes most to the relationship between the metabolic syndrome and oxidative stress. (10.3390/antiox11010079) It is also worth mentioning that oxidative modification of lipoproteins may make them more atherogenic. An example of such modified lipoproteins is nitrated lipoproteins, whose importance in the pathogenesis of cardiovascular disease in diabetic patients has recently been discussed.

Response:

            We added the ideas about atherosclerosis in lines 54-60. Also, we supplemented the concept of oxidative stress in lines 47-50. In lines 48-49, we added a brief information about lipids peroxidation and atherogenesis. Thank you for these suggestions. We also cited the articles suggested by you.

Comment 3:

In describing the methodology (2.2.), the authors used the term: "other drugs that could interfere with the metabolism." Be very careful with that term. In addition to their main mechanism of action, many drugs have additional mechanisms, often not yet fully understood. Please explain exactly what medications the Authors understand by this term.

Response:

            Please, see lines 106-108. Thank you so much for this concern. We made the necessary changes in the methodology.

Comment 4:

The study was conducted from June to December, and the favorable opinion of the bioethics committee was obtained in September of the same year. Please explain.

Response:

There was a mistake in the month of the approval. We corrected it in the manuscript. Please see line 169-170.

Comment 5:

The description of the statistical analysis methodology needs improvement. Describe how the compliance of the distribution of a given variable with the normal distribution was tested. In the case of variables whose distribution is consistent with the normal distribution, the adequate measure of central tendency and dispersion is the arithmetic mean and standard deviation. The median and interquartile range are acceptable parameters for variables whose distribution differs significantly from the normal distribution.

Response:

We corrected the methodology. Please see lines 177-180.

Comment 6:

Sudden cardiac death, arterial stiffness, and subclinical atherosclerosis were not observed and examined according to the methodology and results presented. So, I do not understand why such topics are brought up in the conclusions and at the end of the discussion.

Response:

            Dear doctor, we modified the conclusion.

Comment 7:

The strengths and limitations of the study should be discussed.

Response:

Thank you for this suggestion. We added Section 5 ("Strengths and limitations"). Please see lines 156-166 of the discussion section.

Comment 8:

 The number of references should be increased (currently very small).

Response:

            References increased from 23 to 36. Please, see lines 194-285 for the references list.

Comment 9:

Although English is without huge mistakes and it is understandable, it would be worth performing the language correction by the specialist.

Response:

We appreciate this suggestion. We hired PaperTrue to proofread our manuscript.

Dear reviewer, thank you so much for revising our manuscript. We know your time is precious, but your suggestions improved our work's quality! We are grateful!

Round 2

Reviewer 1 Report

The paper looks much better with all the changes that you have made, and I get your point that its the 1st brazilian study. Yes, this will supplement clinical treatment of patients with obesity or MetS. I approve of this revised paper 

Reviewer 2 Report

The paper has been significantly improved. In my opinion, it represents high scientific quality. I have no further comments. I recommend the manuscript for publication in its current form.